# New Prospects for Improving Microspore Embryogenesis Induction in Highly Recalcitrant Winter Wheat Lines

**DOI:** 10.3390/plants13030363

**Published:** 2024-01-25

**Authors:** Ewa Dubas, Monika Krzewska, Ewa Surówka, Przemysław Kopeć, Agnieszka Springer, Franciszek Janowiak, Dorota Weigt, Sylwia Katarzyna Mikołajczyk, Anna Telk, Iwona Żur

**Affiliations:** 1The Franciszek Górski Institute of Plant Physiology Polish Academy of Sciences, Niezapominajek 21, 30-239 Kraków, Poland; e.dubas@ifr-pan.edu.pl (E.D.); m.krzewska@ifr-pan.edu.pl (M.K.); e.surowka@ifr-pan.edu.pl (E.S.); p.kopec@ifr-pan.edu.pl (P.K.); a.janas@ifr-pan.edu.pl (A.S.); f.janowiak@ifr-pan.edu.pl (F.J.); 2Department of Genetics and Plant Breeding, Poznań University of Life Sciences, 11 Dojazd St., 60-632 Poznań, Poland; dorota.weigt@up.poznan.pl (D.W.); sylwia.mikolajczyk@up.poznan.pl (S.K.M.); 3Department of Analytical Chemistry, Faculty of Chemistry, Jagiellonian University in Kraków, Gronostajowa 2, 30-387 Kraków, Poland; anna.telk@uj.edu.pl

**Keywords:** microspore embryogenesis, winter wheat, macro- and micronutrients, stress, antioxidants, hydrogen peroxide

## Abstract

Among various methods stimulating biological progress, double haploid (DH) technology, which utilizes the process of microspore embryogenesis (ME), is potentially the most effective. However, the process depends on complex interactions between many genetic, physiological and environmental variables, and in many cases, e.g., winter wheat, does not operate with the efficiency required for commercial use. Stress associated with low-temperature treatment, isolation and transfer to in vitro culture has been shown to disturb redox homeostasis and generate relatively high levels of reactive oxygen species (ROS), affecting microspore vitality. The aim of this study was to investigate whether controlled plant growth, specific tiller pre-treatment and culture conditions could improve the potential of microspores to cope with stress and effectively induce ME. To understand the mechanism of the stress response, hydrogen peroxide levels, total activity and the content of the most important low-molecular-weight antioxidants (glutathione and ascorbate), as well as the content of selected macro- (Mg, Ca, NA, K) and micronutrients (Mn, Zn, Fe, Cu, Mo) were determined. These analyses, combined with the cytological characteristics of the microspore suspensions, allowed us to demonstrate that an increased microspore vitality and stronger response to ME induction were associated with higher stress resistance based on more efficient ROS scavenging and nutrient management. It was shown that a modified procedure, combining a low temperature with mannitol and sodium selenate tiller pre-treatment, reduced oxidative stress and improved the effectiveness of ME in winter wheat lines.

## 1. Introduction

Wheat (*Triticum aestivum* L.) is one of the most important crops in the world, covering an approximate harvested area of nearly 221 million ha and a global production volume of 778 million metric tons in the marketing year of 2021/22 (https://www.statista.com/topics/1668/wheat/#topicOverview, URL (accessed on 18 December 2023). It provides approximately 20% of the energy and protein for humans, and serves as an important source of many beneficial compounds such as soluble fiber, vitamins (e.g., thiamine, niacin, folic acid) and minerals (selenium, manganese, phosphorus, copper).

Domesticated around 10,000 years ago at the advent of agriculture, wheat has undergone continuous improvement through selective breeding [1]. Over the past 60 years, the global wheat yield has increased at a rate of about 40 kg/ha/year [2]. Despite this progress, the primary focus of breeding remains on increasing yield, as global wheat consumption is projected to rise by 11% by 2031, according to the Food and Agriculture Organization (FAO) [3]. Ongoing changes in the natural environment and climate, including global warming, pollution and water, further necessitate an increase in yield potential [2]. Other breeding objectives, such as increasing resistance to biotic stresses and improving industrial, nutritional and health properties, are also very important.

Improvements in the wheat quality are possible through biological advances and innovative crop management methods. The former can be facilitated by specific biotechnological techniques, including the incorporation of doubled haploid (DH) lines into breeding programs. DH technology allows for the rapid generation of completely homozygous “true breeding” lines from heterozygous parental genotypes [4]. Its application not only saves time, but also reduces labor input and costs associated with the release of a new cultivar. Of the various methods of DH production, redirecting microspore development towards embryogenesis is considered one of the most efficient. However, developing a procedure for microspore embryogenesis (ME) induction and optimizing in vitro culture conditions are challenging and time-consuming. ME induction requires a complete reprogramming of microspore development and involves many changes in cell structure, metabolism and function. The success of ME process relies on intricate interactions among numerous genetic, physiological and environmental variables. This highly complex interplay is still largely unrecognized and, in many instances, does not function with the efficiency required for commercial use. For example, Polish winter cultivars and bread wheat breeding lines demonstrate low or very low responsiveness to ME [5,6].

It has been shown that an efficient antioxidant defense system is one of the most important factors required for effective microspore reprogramming [7,8,9,10,11]. This mechanism protects microspores from the deleterious effects of the excess generation of reactive oxygen species (ROS). A growing body of evidence also indicates that ROS production is not merely a side effect of stress treatment used for ME induction; rather, it plays a key role in microspore reprogramming and embryo-like structure (ELS) development [10,11,12,13]. Among various ROS, hydrogen peroxide (H_2_O_2_) emerges as a likely candidate for initiating a switch in microspore development due to its moderate reactivity, relatively long half-life and ability to diffuse across cell membranes [9,11,14,15]. The mobility of H_2_O_2_ offers a dual role, acting both as a transducer and an initiator of signaling pathways [14,15].

The aim of this study was to uncover the primary factors contributing to the high recalcitrance of Polish winter wheat breeding materials to ME induction. For this purpose, first, the effectiveness of ME was assessed for several Polish wheat F1 crosses with high breeding value and various origins, employing different culture protocols for isolated microspores and anthers. Subsequently, the effect of several modifications introduced into the ME induction treatment on microspore reprogramming and ELS formation was examined. Utilizing the most promising modified procedure, the changes in the contents of macro- and micronutrients, H_2_O_2_ generation and antioxidant defense provided by low-molecular-weight antioxidant molecules in wheat anthers were monitored. Cytological observations, coupled with the analyses of the obtained data, allowed us to identify the main bottlenecks in the effective induction of microspore reprogramming and haploid/DH plant formation. This newfound knowledge can be instrumental in optimizing standard protocols to overcome barriers to effective ME induction in recalcitrant wheat genotypes.

## 2. Results

Our research began with two experiments conducted during two growing seasons (2020/2021 and 2021/2022), aiming to characterize Polish winter wheat materials in terms of their “embryogenic potential”. The plants grown in the field or under controlled conditions. The protocols used in these experiments were based on standard procedures implemented in our laboratories and an extensive review of published data (e.g., [16,17,18,19,20,21,22]).

### 2.1. Microspore Yield, Viability and Cytological Characteristic (Experiment 2020/2021)

The mean microspore yield of the wheat lines obtained in the first experiment ranged from approx. 14,900 to 54,100 microspores per spike (Figure 1 and Appendix A). Despite a considerable level of variation, the effects of the plant genotype and growth conditions were not significant (*p* > 0.05).

The growth environment of the donor plant exerted a significant effect on the microspore viability (Figure 1). The mean percentage of viable microspores, as measured by FDA staining, was significantly higher for plants grown in controlled conditions compared to field-grown plants. This difference was particularly pronounced for the two more responsive wheat lines (PO19 and PO20), which showed a 45% and 75% increase in microspore viability in suspensions isolated from the field- and greenhouse-grown plants, respectively. In comparison, the mean microspore viability of the five breeding lines of the F1 generation increased by 38% for the field-grown plants and 60% for the plants grown under controlled conditions.

The cytological analysis of the microspores cultured in vitro confirmed a high number of dead or damaged cells in the microspore suspensions on the day of isolation (Figure 2A,B). The microscopic observations (Appendix A) also revealed embryogenic microspores, such as star-like structures (SLSs) and structures after the first symmetric division (2-Nsym), as well as microspores continuing gametophytic development after the first asymmetric division (2-Nasym). However, in the majority of the SLSs, starch grain accumulation was observed in the cytoplasmic strands separating the vacuoles and connecting the perinuclear cytoplasm with the subcortical cytoplasm. The percentage of potentially embryogenic microspores with typical features of developmental reprogramming (SLS + 2-Nsym) varied markedly and depended on the plant genotype, growth conditions and their interaction (Figure 2A,B). For the PO19, PO20, K393 and SM IHAR lines, a visibly higher mean number of SLS + 2-Nsym structures was observed in the microspore suspensions isolated from the donor plants grown under controlled conditions (47%) compared to the field-grown plants (34%). On the contrary, for the other F1 lines, the microspore suspensions isolated from the field-grown donor plants were characterized by a higher ‘embryogenic potential’ (39% versus 32%). Among the cultures started from plants grown under controlled conditions, the number of potentially embryogenic microspores (SLS + 2-Nsym) was highest for PO19 (52%), intermediate for PO20, SM IHAR and K393 (44–46%) and lowest for CH1, K20290 and MHR (30–33%). The same analysis carried out on the isolated microspore cultures started from the field-grown plants showed a more uniform percentage of potentially embryogenic microspores, ranging from 32% to 39%.

The subsequent analyses conducted after 14 and 28 days of culture showed that the majority of the starch-containing SLSs and 2-Nsym died during the first two weeks of in vitro culture. A limited number of reprogrammed microspores divided and formed multicellular structures (Appendix A), which were observed on day 14 of culture. In most cultures, ELS development continued to the globular stage (0.5–1 mm) and then ceased. Cell wall rupture was a commonly observed phenomenon, leading to the leakage of cellular contents and the disruption of the structures (Appendix A). Only a small fraction of the structures reached a size > 1 mm, enabling their transfer to the regeneration medium. Additionally, callus-like structures composed of elongated, loosely arranged cells were also observed after 14 and 28 days of culture (Appendix A). In general, similar to the ELSs, they stopped developing at a stage that was insufficiently advanced to transfer them to the regeneration medium.

### 2.2. Effectiveness of ME Induction in Isolated Microspore and Anther Cultures (Experiment 2020/2021)

Properly developed ELSs were observed in three of the seven wheat lines examined (Appendix A). The highest number of ELSs (16 ELS/spike) was obtained in the PO19 cultures with microspores isolated from the field-grown donor plants. A significantly lower effectiveness was observed for PO20 (2 ELS/spike) and K20290 (0.9 ELS/spike) in the microspore suspensions isolated from the plants grown under controlled conditions. However, despite the successful development of the ELSs, the regeneration process resulted in limited success. No green regenerants were obtained, and only one albino plant was regenerated from the K20290 line.

In the anther cultures, properly developed ELSs were produced by all the wheat lines studied, but only from the donor plants grown in the field, with the effectiveness ranging from 0.7 to 3.2 ELS/100A (Appendix A). Induction on C17 medium supplemented with Ficoll 400 (C17_FIC_) was markedly more effective. However, few ELSs (0.2/100A) were developed in PO20 culture on solid Phytagel-containing medium (C17_PHY_). Green regenerants in the range of 0.3–2 (GR/100A) were obtained for the PO20, SM IHAR and MHR lines. Additionally, albino plants were regenerated in the SM IHAR and PO19 cultures at a frequency of 0.2 AR/100 A.

### 2.3. Effect of Spike Pre-Treatment Modifications on Microspore Yield, Viability and Cytological Characteristics (Experiment 2021/2022)

The prolonged low-temperature tiller pre-treatment had an adverse effect on the microspore yield, which ranged from 900 to 21,100 microspores per spike depending on the genotype (Figure 3). Interestingly, there was no effect of the donor plant growth conditions on the microspore viability, as the difference between the mean viability of the microspores isolated from the field-grown plants (60%) and the plants grown under controlled conditions (65%) was not significant. The mean percentage of viable microspores in the suspensions of the two more responsive wheat lines (PO19 and PO20) reached 72%, which, similarly to the results of the first experiment, was significantly higher compared to the F1 generation breeding lines (60%).

The prolonged low-temperature tiller pre-treatment also exerted some effect on the reprogramming of the microspores (Figure 4A,B). The mean number of potentially embryogenic microspores (SLS + 2 Nsym) increased slightly (from 37 to 42%) in the microspore suspensions of the recalcitrant F1 lines isolated from the donor plants grown under controlled conditions compared to the first experiment. This effect was neither observed for the two more responsive lines, PO19 and PO20, nor in the suspensions isolated from the field-grown plants. Conversely, the number of SLS + 2 Nsym structures decreased in most of these suspensions. As a result, the percentage of SLS + 2 Nsym structures in the microspore cultures isolated from the plants grown under controlled conditions was highest in K393 (70%), intermediate in PO20, PO19 and SM IHAR (43–52%) and lowest in MHR, CH1 and K20290 (22–37%). Once again, as in the first experiment, the same analysis carried out on the microspore cultures started from the field-grown plants showed a more uniform percentage of potentially embryogenic microspores, ranging from 27% to 40%.

Similar to the previous experiment, the majority of these potentially embryogenic structures continued their development to the multicellular or globular stage, and subsequently went into cell cycle arrest.

Modifications to the ME induction procedure were subsequently applied to four wheat lines that differed highly in their responsiveness to the ME induction procedure (K20290, SM IHAR, PO19 and PO20).

The low-temperature pre-treatment (21–28 days at 4 °C) was applied as a single stress factor and exerted a positive effect on the microspore yield and viability in three wheat lines, namely K20290, SM IHAR and PO20 (Figure 5). However, this beneficial effect was not recorded in PO19. The observations made on day 14 of the in vitro culture showed that almost all the microspores were non-viable, irrespective of the wheat genotype.

The combination of the low- and high-temperature tiller pre-treatment (21–28 days at 4 °C, followed by 24 h at 30 °C) proved lethal for the microspores of the K20290 and SM IHAR lines, as no viable microspores were collected after density gradient centrifugation (Figure 5). Conversely, the same pre-treatment resulted in high to very high microspore yields, as well as a high microspore viability in the PO19 and PO20 lines. Moderate to high numbers of SLSs were observed in the isolated microspore cultures of the PO19 and PO20 lines, respectively. However, their development was arrested at the multicellular or globular stage. Individual properly developed ELSs and callus-like structures were observed only in the cultures of the PO20 line.

The low temperature combined with the 0.4 mol/dm^3^ mannitol treatment resulted in a relatively low microspore yield compared to the other modifications (Figure 5). The microspore viability varied between 48% and 67% for SM IHAR and K20290, while PO19 and PO20 exhibited viability at 90%. Numerous SLSs were observed in all the cultures, but their development was arrested around the multicellular stage.

Among the various ME induction variants of the tiller pre-treatments, the sequence involving low temperature, SSe and 0.7 µmol/dm^3^ mannitol at 20 °C proved the most promising. In general, this procedure resulted in a moderate microspore yield and high microspore viability (Figure 6), but the effect of this modification was genotype- and SSe-dose dependent. Pre-treatment with 50 µmol/dm^3^ SSe was particularly beneficial for the two more responsive wheat lines, PO19 and PO20, resulting in high microspore yields (from 20,200 to 33,600 microspores per spike) and high cell viability (74–92%). Most importantly, this pre-treatment resulted in a high number of SLSs (41–69%) along with a very low number of 2-Nsym (0.9–1.9%; Figure 6). A higher dose of SSe (100 µmol/dm^3^) resulted in a lower microspore yield (Figure 6) and a reduced percentage of the occurrence of reprogrammed microspores (31–42% SLS and 0.8–1% 2-Nsym; Figure 6) in PO19 and PO20. The microspore yield and viability increased with a higher SSe concentration in both the K20290 and SM IHAR lines (Figure 6). However, the percentage of ME induction was lower compared to the more responsive wheat lines (21–31% SLS + 2-Nsym for 50 µmol/dm^3^ SSe and 19–28% for 50 µmol/dm^3^ SSe).

### 2.4. Effectiveness of ME Induction in Isolated Microspore and Anther Cultures (Experiment 2021/2022)

The ME induction procedure, which combines an extended low-temperature tiller pre-treatment with a short high-temperature shock, and the application of 0.4 mol/dm^3^ mannitol, resulted in the generation of single, properly developed ELSs in isolated microspore cultures of the K393 and PO19 lines (Appendix A). The effectiveness of ELS generation from the plants grown under control conditions was 0.7 ELS/spike and from the field grown PO19 plants of 0.1 ELS/spike. Green regenerants (0.2 GR/spike) were obtained only for the P019 donor plants grown under controlled conditions.

Among all the other modifications introduced to the ME induction procedure, only the pre-treatment combining low temperature with SSe and 0.7 mol/dm^3^ mannitol maintained the proper development of ELSs in two wheat lines, i.e., PO19 (1.1 ELS/spike) and PO20 (2.7 ELS/spike). The regeneration of green plants was observed in the cultures started from the field-grown plants of the PO19 (0.2 GR/spike) and PO20 (0.6 GR/spike) lines.

In the anther cultures, a 10-day pre-treatment of the tillers with low temperature and using media supplemented with Ficoll 400 resulted in ELS formation in six of the seven wheat genotypes studied, with an effectiveness ranging from 0.6 to 4 ELS/100 A (Appendix A). ELS production was also observed on C17 medium solidified with Phytagel for three F1 lines (CH1, K20290 and MHR), with an effectiveness of 0.6 ELS/100 A. After 20 days of low-temperature tiller pre-treatment, four winter wheat lines (CH1, K29290, PO19 and PO20) generated ELSs on Ficoll-supplemented medium with an effectiveness ranging from 0.6 to 2.6 ELS/100 A. Similarly as before, a lower effectiveness of ME induction was observed on the Phytagel-solidified C17 medium (2 ELS/100 A for two winter wheat lines). Only one wheat line (CH1) produced ELSs after low-temperature tiller pre-treatment extended to 30 days, with an effectiveness of 0.6 ELS/100 A. For one wheat line (K393), no ELS development was observed regardless of the length of the tiller pre-treatment and type of culture medium.

On the basis of the results, the tiller pre-treatment procedure combining low temperature, SSe and 0.7 mol/dm^3^ mannitol was used to study several physiological parameters characterizing the anthers of three wheat lines: two highly recalcitrant K20290 and SM IHAR, and a more responsive PO19.

### 2.5. Macro- and Micronutrient Contents in Wheat Anthers and Their Changes Induced by Modified ME Induction Procedure

Among the mineral macronutrients analyzed, magnesium (Mg) and calcium (Ca) accumulated at relatively high contents in the anthers collected from the non-pre-treated tillers (control) of all the wheat lines studied. Mg was detected at about 2.5 mg/g dry weight (DW) and Ca in a range from 1.6 to approx. 2 mg/g DW (Figure 7). The content of sodium (Na) was more than 10-fold lower (65–130 µg/g DW), while the potassium (K) content did not exceed 20 µg/g DW. With only one exception (K in PO19 anthers), the content of all four elements was significantly higher in the anthers collected from the plants, the tillers of which were pre-treated with low temperature. It was also observed that the difference between the control and low-temperature treated anthers tended to be higher in the K20290 and SM IHAR lines compared to the PO19 line. The effect of the introduced modification in the tiller pre-treatment procedure was genotype-dependent. Compared to the effect of low temperature, the combined low-temperature, SSe and mannitol pre-treatment increased (K20290) or decreased (SM IHAR, PO19) the Mg content, increased (PO19) or decreased (K20290) the K content, elevated (K20290) or did not change (SM IHAR, PO19) the Na content, and elevated (SM IHAR, PO19) or did not change (K20290) the Ca content (Figure 7).

The pattern of changes observed for micronutrient minerals was much more consistent (Figure 8). The concentrations of three micronutrients (zinc (Zn), iron (Fe) and manganese (Mn)) ranged from approx. 60 to 100 µg/g DW in the anthers collected from the non-pre-treated tillers. Less abundant was copper (Cu, in the range of 11 to 17 µg/g DW, and Mo, detected at 0.2 0.4 µg/g DW).

Low temperature increased Zn accumulation in the anthers of all the wheat genotypes studied. It also increased the contents of Fe in the anthers of the K20290 and SM IHAR lines, Mn in the anthers of the K20290 line, and Cu in the anthers of the SM IHAR and PO19 lines. On the contrary, the content of Mo significantly decreased in the anthers derived from the tillers pre-treated with low temperature in the K20290 and SM IHAR lines.

The modified pre-treatment of the tillers with low temperature, SSe and mannitol did not induce any further changes in the PO19 anthers. The content of Fe in the SM IHAR line was significantly reduced compared to the anthers from the low-temperature pre-treated tillers. Significantly greater differences were observed in the K20290 anthers, where the Fe, Mn and Cu contents were significantly reduced, while the Mo content was significantly increased in response to the modified ME induction pre-treatment.

As a result of all the changes induced by the modified tiller pre-treatment, the PO19 anthers were characterized by lower contents of two mineral elements (Mg and Fe) in comparison to SM IHAR. The same comparison with the K20290 line showed lower contents of four micronutrients (Mg, Na, Fe and Mo) and a higher content of Cu.

### 2.6. Glutathione and Ascorbate Levels and Redox Balance in Wheat Anthers and Changes in These Parameters Induced by the ME Induction Procedure

The level of total glutathione measured in the anthers collected from the non-pre-treated tillers of the tested wheat lines ranged from 0.35 to 0.7 µmol/g fresh weight (FW; Table 1). The glutathione concentration was highest in the anthers of the K20290 line and lowest in the PO19 line. Similarly, GSH accumulation (Figure 9a) was highest in the K20290 line (0.5 µmol/g FW), slightly lower in the anthers of the SM IHAR line (0.43 µmol/g FW) and two times lower in the PO19 anthers (0.25 µmol/g FW). All the anthers also contained relatively low levels of GSSG (0.1–0.2 µmol/g FW; Figure 9b). The proportion of GSH in the total glutathione pool ranged from 72% in the K20290 and PO19 anthers to 85% in the SM IHAR anthers (Table 1).

The anthers of the K29290 line, collected after low-temperature tiller pre-treatment, accumulated a slightly higher (by 6%) GSH content and significantly lower (by 40%) GSSG content (Figure 9a,b). On the contrary, reduced GSH contents (by 47% and 39%) were observed in the SM IHAR and PO19 anthers, respectively. This was accompanied by an increased quantity of GSSG, especially in the SM IHAR line, where its level was 8.5-fold higher compared to the anthers collected from the non-pre-treated tillers. The level of total glutathione (Table 1) decreased in the K20290 anthers (to 0.65 µmol/g FW), slightly increased in the PO19 anthers (to 0.37 µmol/g FW) and rose sharply in the SM IHAR anthers (to 0.89 µmol/g FW) compared to the corresponding control. The GSH pool in the total glutathione increased to 82% in the K20290, but decreased to 26% and 41% in the SM IHAR and PO19 anthers, respectively.

The response to the combined pre-treatment with low temperature, SSe and mannitol was also genotype-dependent (Figure 9a,b). Compared to the tillers pre-treated with low temperature alone, a lower amount of GSH was determined in the K20290 anthers, but its increased accumulation was recorded in the SM IHAR and PO19 anthers. Interestingly, the levels of GSSG and total glutathione drastically decreased in the anthers of all the wheat lines studied compared to the corresponding control and low-temperature pre-treated anthers (Figure 9b, Table 1). The GSH level reached approx. 95% of the total glutathione level in the K20290 and SM IHAR anthers and 87% in the PO19 anthers.

The accumulation of total ascorbate in the wheat anthers ranged from 0.8 to 2.2 µmol/g FW and was on average 3.5 times higher than that of glutathione. Similarly to glutathione, the levels of the reduced and oxidized forms of ascorbate measured in the isolated anthers varied significantly among the wheat lines and tiller pre-treatments studied (Figure 9c,d).

The ASC levels in the anthers collected from the non-treated tillers were the highest in the PO19 line (1.3 µmol/g FW) and markedly lower in the anthers from the K20290 (0.4 µmol/g FW) and SM IHAR (0.1 µmol/g FW) lines (Figure 9c). Higher ASC levels were associated with lower DHA accumulation (Figure 9d), and the ratio of ASC to the total ascorbate pool ranged from 5% in the SM IHAR anthers to 66% in the PO19 anthers (Table 1).

The low-temperature tiller pre-treatment increased the ASC accumulation (Figure 9c) in the anthers of two wheat lines: K20290 (to 0.6 µmol/g FW) and SM IHAR (to 1.5 µmol/g FW). The same pre-treatment slightly reduced the ASC accumulation in the anthers of the PO19 line. These changes were associated with lower DHA accumulation in the K20290 and SM IHAR anthers (Figure 9d). A different effect was once more observed for the PO19 line (an increase from 0.7 to 0.9 µmol/g FW). The ratio of ASC to the total ascorbate pool (Table 1) increased greatly in the SM IHAR anthers (from 5 to 76%), moderately in the K20290 anthers (from 24 to 38%) and decreased in the PO19 anthers (from 66 to 59%).

In all the wheat lines studied, the combined low-temperature/SSe/mannitol pre-treatment reduced the ASC levels to values below 0.4 µmol/g FW (Figure 9c). Additionally, the procedure resulted in reduced DHA accumulation compared to the anthers from the low-temperature pre-treated tillers (Figure 9d); its content, as well as the content of ASC, were highest in the anthers of the PO19 line. This effect was also observed in the level of total ascorbate, which decreased drastically after this modified tiller pre-treatment (Table 1). The proportion of ASC in the total ascorbate pool was similar in the PO19 and K20290 anthers (30–33%) and relatively higher (43%) in the SM IHAR anthers.

### 2.7. Total Activity of Low-Molecular-Weight Antioxidants in Wheat Anthers and Alterations in This Activity Stimulated by the ME Induction Procedure

The total activity of low-molecular-weight antioxidants measured in the K20290 and PO19 anthers collected from the non-pre-treated tillers amounted to 1.3 µmol Trolox equivalents/g DW (Figure 9e). It was significantly higher in the anthers of the SM IHAR line (1.9 µmol Trolox equivalents/g DW).

In the anthers of the two highly recalcitrant lines, K20290 and SM IHAR, collected from tillers treated with low temperature, the total activity of low-molecular-weight antioxidants decreased to 1.1–1.4 µmol Trolox equivalents/g DW, whereas it remained unchanged in the PO19 anthers.

In all the studied wheat lines, the antioxidant activity increased significantly after the modified ME induction procedure combining low temperature, SSe and mannitol; however, the amplitudes of these changes were visibly different. In comparison with the low-temperature tiller pre-treatment, the procedure increased the antioxidant activity more than 2-fold in the K20290 and SM IHAR anthers, and more than 3-fold in the PO19 anthers. Thus, the final ROS detoxification capacity of the low-molecular-weight antioxidant molecules was 1.5 to almost 2 times higher in the anthers of the more responsive PO19 line compared to the recalcitrant lines, SM IHAR and K20290, respectively.

### 2.8. H_2_O_2_ Generation in Wheat Anthers and Changes in this Parameter Caused by the ME Induction Procedure

The level of H_2_O_2_ generated in the anthers of the studied wheat lines collected from the non-pre-treated tillers ranged from 23.9 to 28.5 nmol/g FW, and was highest in the K20290 line, intermediate in PO19 and lowest in the SM IHAR line (Figure 9f).

It was significantly reduced (by 17–42%) in the anthers of the SM IHAR and PO19 lines collected from the tillers pre-treated with low temperature, while it was unchanged in the K20290 anthers.

A decrease in the H_2_O_2_ level was observed in all the wheat lines tested after applying the combined tiller pre-treatment with low temperature, SSe and mannitol. This parameter exhibited a significant decrease in the anthers of the more responsive line PO19 (to 34% of the control and 58% of the value measured after low-temperature tiller pre-treatment) and a moderate decrease in the anthers of the SM IHAR and K20290 lines (to 59–66% of the control and about 70% of the value measured after the low-temperature tiller pre-treatment).

## 3. Discussion

This research sheds new light on the possibilities of incorporating DH technology in wheat breeding. While confirming the high recalcitrance of Polish winter wheat lines to ME induction, this study also explored genotypes with potentially higher responsiveness (PO19 and PO20), selected on the basis of previous studies.

Realizing this potential required specific environmental and procedural conditions. The comparison between these potentially responsive and highly recalcitrant lines in terms of parameters characterizing the microspore condition, stress intensity and defense reactions, allowed us to indicate the possible determinants of low responsiveness to ME induction. Having a unique set of genotypes, the study could identify the physiological basis of recalcitrance to ME induction and propose promising protocols to enhance microspore viability by increasing tolerance to oxidative stress associated with isolation and transfer to in vitro culture.

### 3.1. Influence of Donor Plant Growth Conditions, Tiller Pre-Treatment and In Vitro Culture Technique on the Efficiency of Wheat ME

Although the average microspore yield (approx. 28,600 microspores per spike) obtained by mechanical isolation was low, this does not seem to be a critical factor, as similar microspore yields were typically obtained for the highly embryogenic barley cultivar Igri [13]. Challenges begin when the low number of microspores produced is coupled with low cell viability, as observed in the investigated winter wheat lines. This problem mainly concerns plants grown under natural conditions, where weather conditions play a significant role. In wheat, high temperatures are particularly dangerous, as they can lead to the premature degeneration of the tapetum and significantly accelerate microsporogenesis [23]. This problem was observed in the first experiment (2020/2021) when the average percentage of viable, metabolically active microspores was about 40%. In comparison, the initial microspore viability of the responsive wheat cultivar Pavon 76 was approx. 75% [24]. The pollen developmental stages in the peri-meiotic period are known to be highly stress sensitive, often leading to microspore abortion and male sterility [25]. Depending on the stress factor and its duration, cellular effects can include the modulation of meiotic recombination, cytoskeletal rearrangements, changes in sugar and hormone metabolism, epigenetic modifications or the induction of programmed cell death [26]. The fact that a low number of viable cells was also observed in the microspore suspensions isolated from the plants grown under controlled conditions suggests a generally high sensitivity to stress associated with isolation and transfer to in vitro culture. On the other hand, the surviving microspores isolated from the field-grown plants exhibited a higher potential for proper ELS formation compared to microspores produced under controlled conditions. This effect was observed in both the isolated microspores and anther cultures. Similarly, Dogramaci-Altuntepe et al. [27] reported that ME induction and green plant regeneration were more effective in field-grown plants compared to plants cultivated in controlled growth chambers.

The low viability of the microspores was not the only reason for the low effectiveness of the ME induction. According to numerous reports, wheat genotypes can effectively induce ME in response to low temperature or to a combination of heat stress with carbohydrate and nitrogen starvation [16,17,18,19,24]. However, no such effect was observed for the winter wheat lines examined in the present study. The effectiveness of the microspore reprogramming was low and accompanied by severe disturbances in ELS development. Low temperature in combination with mannitol and high temperature resulted in intensive starch accumulation, also found in the SLS and 2-Nsym structures. This observation suggests that the microspores were not completely redirected from the gamethophytic developmental pathway, as previously reported by Indrianto et al. [28] and Hu and Kasha [29]. This was further confirmed by the results of the following experiment involving prolonged low-temperature tiller pre-treatment (21–28 days at 4 °C). The latter modification was introduced in an attempt to increase the stress tolerance of the microspores and was based on published data indicating that low temperatures could induce a variety of defense responses, thereby increasing the rate of stress survival by microspores (reviewed by Zoriniants et al. [30]). This effect was observed in our previous studies involving triticale, where a low-temperature tiller pre-treatment increased the microspore yield and viability, as well as enhanced the abscisic acid accumulation and respiration rate [31]. In the subsequent study, the induction of antioxidant defense and reduced ROS generation were detected in microspores exposed to low temperature [9].

The present study confirmed the positive effect of prolonged low-temperature treatment on microspore viability, but it was associated with the progression of gametophytic development and ultimately the low efficiency of ME induction. The fact that a much shorter (10–14 d) period of low-temperature pretreatment induced, to some extent, ELS formation in the anther cultures suggests a role of the anther tapetum in stress alleviation and microspore reprogramming. In plants, the tapetum provides microspores with nutrients and enzymes required for pollen development [32]. It synthesizes precursors for pollen exine formation and accumulates them until the onset of programmed cell death (PCD), when these pollen wall components are transferred into pollen grains [33,34]. In addition, according to a recent study [32], the tapetum can provide microspores with gene expression regulators (microRNAs and small interfering RNAs). Anther tissues are also a site of auxin accumulation, which is crucial for cell proliferation and differentiation. In triticale, indole-3 acetic acid was detected in the tapetum of the anther–filament junction and in the protoxylem of the procambial strand [35]. It could be assumed that the absence of such supplements could lead to structural and/or functional abnormalities in ELSs developing in isolated microspore culture.

The effect of the introduced modifications to the ME induction procedure, especially the combination of low temperature and heat shock, on microspore viability also suggests that recalcitrant winter wheat lines are more stress sensitive compared to lines showing higher responsiveness. This also indicates disturbances in the following stages of embryogenic development, given that even more intense stress (4 days at 33 °C) applied to tillers of other winter wheat cultivars was shown to be highly effective in inducing ME and ELS development [16]. Based on our previous studies in recalcitrant rye (*Secale cereale* L.), the application of SSe was used as an additional method to increase the stress tolerance of wheat microspores [36]. Although selenium (Se) is a trace element, it is essential for animals and some microorganisms, and exerts beneficial effects on the growth and development of higher plants [37]. Plants primarily absorb Se in inorganic forms such as selenate or selenite. Due to its chemical similarity to sulfur, it is taken up by sulfate transporters (SULTRs) located in plasma membranes and readily transported from plant roots to shoots and leaves. In plant cells, Se is incorporated into biologically active organic forms of Se amino acids, i.e., selenocysteine and selenomethionine, which are used for selenoprotein synthesis [38]. At low concentrations, Se enhances photosynthesis and increases the accumulation of carbohydrates and secondary metabolites [39]. It is also involved in antioxidant defense, enhancing the activity of superoxide dismutase (SOD), catalase (CAT), ascorbate peroxidase (APX), glutathione reductase (GR) and glutathione peroxidase [40,41]. Increased stress tolerance following Se application has been observed in various plant species and stress conditions, such as cold, drought, desiccation or heavy metals [37].

In the pursuit of achieving enhanced Se stress tolerance in the same pre-treatment procedure, the stress intensity was increased by prolonged treatment with mannitol at a higher concentration (4 days in 0.7 mol/dm^3^ mannitol). Mannitol induces symmetric microspore division, which has been described as the most efficient for ELS production in wheat ME [42]. A similar treatment used by Cistué et al. [20] proved effective in ME induction and plant regeneration in highly recalcitrant durum wheat. While this treatment showed promise also in our study, green plant regeneration was only observed in more responsive wheat lines (PO19 and PO20). Once again, as demonstrated earlier, the presence of SLSs and even the initiation of symmetric divisions did not guarantee successful ME induction. The frequently observed abortion of embryogenic structures at the multicellular or globular stage of development suggests disruptions in the molecular control or metabolic pathways necessary for proper ELS formation.

### 3.2. Effect of ME Induction Procedure on Micro- and Macronutrient Levels, Intensity of Oxidative Stress and Non-Enzymatic Antioxidant Defense in Wheat Anthers

The wheat anthers of three breeding lines (K20290, SM IHAR and PO19) were analyzed for the content of selected micro- and macronutrients, ascorbate, glutathione, total activity of non-enzymatic antioxidants and H_2_O_2_ generation during the modified ME induction procedure. The analyses were carried out in order to (i) determine the effect of the low-temperature tiller pre-treatment, which is a standard procedure for efficient ME induction in different species, and (ii) reveal the response elicited by the combination of low temperature, SSe and mannitol, recognized as potentially the most effective treatment to induce ME in the winter wheat lines studied.

Overall, our study showed fairly uniform contents of the analyzed micro- and macronutrients in the anthers isolated from the non-pre-treated tillers of the winter wheat lines. Some differences observed in the contents of Ca, Na, Fe and Cu were genotype-dependent and not related to their ME induction potential. The only marked difference between the two highly recalcitrant and more responsive wheat lines was a significantly lower accumulation of Mo detected in the control anthers of the PO19 line. As a cofactor of specific enzymes, Mo plays a crucial role in multiple physiological processes regulating plant growth and development. Lower Mo accumulation could indicate the less intensive metabolism of nitrogen-, carbon- and sulfur-containing amino acids [43,44]. Interestingly, the Mo content did not decrease at low temperature only in the responsive PO19 line. Possibly, Mo-containing enzymes, involved in indole-3 acetic acid (IAA) and abscisic acid (ABA) synthesis [45], might regulate the adaptability of microspores to low temperature through changes in endogenous hormone levels. Furthermore, specific hormonal homeostasis and auxin–cytokinin–ABA crosstalk are more important requirements for effective ME than the levels of individual hormones [35].

The fact that the anthers of highly recalcitrant wheat lines (K20290 and SM IHAR) subjected to stress induced by the isolation procedure accumulate more glutathione and much less ascorbate, compared to the more responsive line (PO19), suggests that ascorbate is more important for the stress survival of wheat microspores. Both glutathione and ascorbate belong to highly potent, abundant and stable non-enzymatic antioxidants, and their functions have been extensively reviewed in the literature (e.g., [46,47,48,49,50]). In living cells, under physiological conditions, they predominantly occur in reduced forms (GSH and ASC), which are oxidized in redox reactions to GSSG and DHA. The total pool of glutathione (GSH + GSSG) and ascorbate (ASC + DHA) is responsible for the cellular redox buffering capacity, while the proportion of the reduced form in their total pool (GSH/GSH + GSSG and ASC/ASC + DHA) determines the cellular redox potential, which affects gene expression, signaling and the accumulation of redox-responsive proteins [51]. Both ASC and GSH can act directly as ROS scavengers or as cofactors of certain antioxidant enzymes, e.g., peroxidases. They are involved in the biosynthesis or interact with plant hormones, such as abscisic acid, auxins, gibberellins, jasmonic acid or ethylene, and play a crucial role in plant stress defense (reviewed in [48,50,52]). These two antioxidant molecules act in a compensatory manner, but some of their functions are not interchangeable [46,53]. For example, the apoplast ASC fraction is believed to play a crucial role in oxidative stress signaling (reviewed in [47]). All of the above processes can exert a significant impact on ELS formation. A positive influence of GSH and ASC on ELS development and plant regeneration has been repeatedly described [21,54,55,56,57,58,59]. Their potential role in microspore embryogenesis has also been suggested in our previous studies [10,11,13].

Among the various ROS, H_2_O_2_ appears to be of the most interest as a major redox signaling molecule [60], regulating plant growth, development and stress defense [61,62]. Increasing evidence also points to its involvement in microspore reprogramming and differentiation [9,11,13]. The relatively high level of H_2_O_2_, together with the low overall activity of the antioxidant system, observed in the anthers isolated from the tillers without any pre-treatment, have confirmed that the isolation procedure is associated with the intense accumulation of ROS, which could adversely affect the microspore viability. It has been shown that excessive ROS accumulation can cause cytoplasmic male sterility [25].

The low temperature induced significant changes in nutrient management. It stimulated the accumulation of the analyzed macro- and micronutrients (except Mo), with significantly stronger effects observed in the highly recalcitrant wheat lines (K20290 and SM IHAR). The increased accumulation of Mg, K and Fe, coupled with an extreme reduction in the Mo content, indicated the disruption of cellular homeostasis and extensive changes in cellular metabolism. The effects of low-temperature stress on wheat plant mineral contents is rather poorly recognized [63]. The similarity between temperature-induced changes, i.e., the elevation of the Ca, Mg, Na, Cu and Zn content, but the decrease in the Mo content after the low-temperature pre-treatment, was also previously observed in cauliflower seedlings by Kalisz et al. [64]. The data obtained indicate the initiation of cell defense responses under temperature stress. The increase in the K content may be related to its ability to lower the water potential in cells [65]. Beyond osmoregulation, K controls the electrical potential of cell membranes and stabilizes their structure at low temperatures, modifies intracellular pH and activates enzymes [66,67]. For example, a higher content of K have been found in drought-resistant spring wheat cultivars compared to drought-sensitive cultivars [68].

Similarly, the content of Ca, a secondary messenger involved in the transmission of cellular information, is closely associated with stress-resistance in plants [69]. Elevated contents of Zn and Cu, which are essential co-factors for many enzymes (including superoxide dismutase and catalase), can enhance the cells’ capacity to cope with oxidative stress. The increase in the Fe content may be attributed to the cold-induced expression of genes related to Fe uptake [70].

A relatively weaker effect of low temperature on the contents of macro- and micronutrients was observed in the PO19 anthers, which, together with a strong reduction in H_2_O_2_ generation, indicates that this wheat genotype has a relatively higher tolerance to cold stress. In this case, low temperature can be considered a factor that triggers various stress defense mechanisms, as suggested by Zoriniants et al. [30]. For example, the activity of catalase and non-specific peroxidase under low temperature was shown to determine the high responsiveness of triticale DH lines to ME induction [9]. The observation that even a significant increase in ASC accumulation observed after the low-temperature treatment in the SM IHAR anthers had only a limited effect on the H_2_O_2_ levels suggests either increased ROS production or the decreased activity of other elements of antioxidant defense. A similar conclusion can be drawn from the fact that despite the increased content of GSH and ASC, high levels of H_2_O_2_ were detected in the anthers isolated from the low-temperature pre-treated tillers of K20290. All these observations suggest a low tolerance to cold stress of both the recalcitrant wheat lines.

The modified tiller pre-treatment, combining low temperature, SSe and mannitol, stimulated antioxidant defense and significantly reduced H_2_O_2_ production.

The results of this experiment confirmed the highly enhanced antioxidant defense in the anthers subjected to the modified ME induction procedure. This effect was most pronounced in the more responsive PO19 line. Interestingly, it was associated with relatively low levels of both total and reduced ascorbate and glutathione. Since this effect was associated with a substantial increase in the total activity of low-molecular-weight antioxidants, it suggested the involvement of other yet unrecognized non-enzymatic molecules with high efficiency in ROS scavenging.

Several studies have demonstrated the positive effects of Se on plants exposed to various stressors [71,72,73]. Se application affects the accumulation of elements in plants, and this phenomenon is not only dependent on the plant genotype, but also related to the plant development stage and the applied Se dose [74,75]. This has been confirmed by the results of our experiment, which indicate that Se, especially at low concentrations, has a positive influence on the viability and reprogramming of microspores. Low Se concentrations have been shown to stimulate the activity of antioxidant enzymes involved in ROS scavenging [39,41], and thus reduce toxic effects induced by biotic and abiotic stressors [76,77]. However, when using Se as a biostimulant, it is important to note that its hyperaccumulation is toxic and may induce ROS generation [78]. Additionally, it has been reported that seleno-amino acids can be nonspecifically incorporated into proteins, leading to their misfolding and loss of activity [38].

The modified tiller pre-treatment procedure also induced alteration in the mineral composition of the anthers. The increase in the Zn content in all the lines may indicate the additional synthesis (compared to the effects of low temperature) of antioxidant enzymes containing this element to minimize stress effects. The higher Mo accumulation, especially after a significant reduction in the content of this element during the low-temperature pre-treatment, may also be related to the increase in the Mo-containing enzymes. The elevated content of this element has also been observed in Se-treated wheat seedlings [76]. Among the observed changes, an elevated K content and a relatively low Fe content, characteristic of the more responsive PO19 line, appear to be particularly important. Fe is involved in many metabolic processes, such as DNA synthesis, respiration, photosynthesis, nitrogen assimilation or stress defense [79,80]. It is also a component of the prosthetic group of many enzymes involved in redox reactions (cytochromes, ferredoxins, nitrite reductase, etc.), including those engaged in ROS scavenging (superoxide dismutase, catalase and peroxidases). However, the high cellular accumulation of Fe can induce oxidative stress due to the increased generation of hydroxyl radicals, the most reactive and dangerous form of ROS, produced via the Fenton reaction [61].

## 4. Materials and Methods

### 4.1. Plant Material

The five F1 crosses of common winter wheat (*Triticum aestivum* L.) used in the study were acquired from Polish breeding companies: Danko Hodowla Roślin Sp. z o.o. (CH1), Poznańska Hodowla Roślin Sp. z o.o. (K20290), Strzelce Hodowla Roślin Sp. z o.o. IHAR Group (K393), Smolice Hodowla Roślin Sp. z o.o. IHAR Group (SM IHAR) and Małopolska Hodowla Roślin Sp. z o.o. (MHR). Two other winter wheat breeding lines (PO19 and PO20) from the F4 generation were obtained from Danko Hodowla Roślin Sp. z o.o. and identified in preliminary studies as potentially more responsive to ME induction treatment.

### 4.2. Plant Growth

Plants were grown in the field or in a greenhouse. Seeds were sown in mid October at a depth of 5 cm in 20 cm rows. Plants were fertilized with Azofoska (1 N:0.5 P_2_O_5_:1.4 K_2_O) twice: before germination and at the beginning of tillering, in doses recommended by the manufacturer.

The remaining seeds were germinated for 2 days in perlite moistened with Hoagland’s salt solution (HS), as described by Wędzony (2003), and then vernalized at 4 °C with a photoperiod of 8/16 h (day/night) for 7 weeks. The plantlets were subsequently transferred to a mixture of soil and sand (3/1; *v*/*v*) and grown in a greenhouse at 20 °C with a photoperiod of 16/8 h (day/night). Additional illumination (400 µmol m^−2^ s^−1^) was provided by high-pressure sodium (HPS) lamps SON-T+ AGRO (Philips, Brussels, Belgium) during unfavorable weather conditions.

The experiments were carried out in two growing seasons, 2020/2021 and 2021/2022.

### 4.3. Tiller Pre-Treatment for Microspore Embryogenesis Induction

Tillers were harvested when the microspores in the central part of the spike were highly vacuolated during the mid- to late-uninucleate stage, as assessed by 4′,6-diamidine-2′-phenylindole (DAPI) staining. Morphological features of tillers correlated with the optimal stage of microspore development were determined individually for each wheat breeding line.

The collected tillers were defoliated (except for the flag leaf), placed in plastic bags, wrapped in aluminum foil and kept in glass jars containing tap water.

In the first experiment involving the isolated microspore culture method, tillers were incubated at 4 °C in the dark for 14–21 days, then transferred to a 0.4 mol dm^−3^ mannitol solution and incubated at 30 °C for 1–2 days (LT/HT + MAN).

For anther culture, microspore reprogramming was initiated by a standard tiller pre-treatment at low temperature (4 °C) for 10–14 days.

### 4.4. Tiller Pre-Treatment Modifications

Based on the results obtained in the first season, wheat tillers of all tested wheat lines were harvested in the following season (2021/2022) when most microspores were in the mid-uninucleate stage (DAPI staining). For isolated microspore cultures, low-temperature tiller pre-treatment was extended to 21–28 days at 4 °C. For anther cultures, low-temperature pre-treatment of tiller lasted for 10, 20 or 30 days.

Four wheat lines (K20290, SM IHAR, PO19 and PO20) were then subjected to the testing of four modified variants of tiller pretreatment for ME induction: (i)Low-temperature tiller pre-treatment for 21–28 days at 4 °C;(ii)Low/high-temperature tiller pre-treatment for 21–28 days at 4 °C, followed by 24 h at 30 °C;(iii)Low-temperature/osmotic/starvation tiller pre-treatment for 21–28 days at 4 °C, followed by 4 days in 0.4 mol/dm^3^ mannitol at 20 °C;(iv)Low-temperature/osmotic/starvation/selenium tiller pre-treatment for 21–28 days at 4 °C with the last three days incubated in 50 µmol/dm^3^ or 100 µmol/dm^3^ sodium selenate (Na_2_SeO_4_; SSe) solution, followed by 4 days in 0.7 mol/dm^3^ mannitol at 20 °C.

### 4.5. Isolated Microspore Culture

Pre-treated spikes were sprayed with 96% ethanol, surface sterilized in a 20% solution of commercial bleach (‘Domestos’) for 15 min, and rinsed 4–5 times with sterile deionized water. In the next experimental season, the procedure was supplemented by the treatment with 0.005% HgCl_2_ for 1 min, applied after ethanol disinfection due to the increased frequency of culture contamination.

Microspores were mechanically isolated in 0.3 mol/dm^3^ mannitol using a Waring blender (Fisher Scientific Inc., Göteborg, Sweden), followed by filtration through a 74 μm metal sieve (200 mesh; CD-1, Sigma-Aldrich, St. Louis, MO, USA) and pelleted (100× *g*, 4 min). Subsequently, microspores were collected using density gradient centrifugation (0.3 mol/dm^3^ mannitol/21% maltose, 80× *g*, 4 min), washed in 0.3 mol/dm^3^ mannitol and pelleted again (100× *g*, 5 min). Finally, the pellet was resuspended in 1 mL of KBP medium containing 0.9 mg/dm^3^ BAP (Kumlehn et al., 2006 [81]). In the second replication of the experiment, KBP medium was supplemented with 50 mg/dm^3^ arabinogalactan-proteins (AGPs) and 1 mg/dm^3^ 2,4-D.

The total number of washed microspores was estimated using a Neubauer counting chamber. KBP medium was added to obtain a final culture density of 80,000 microspores per cm^3^ (mcs cm^−3^). Then, the suspensions were transferred to 15 × 60 mm Petri dishes and co-cultured with immature ovaries (10 per 1.5 mL of suspension), which had been dissected simultaneously with microspore isolation procedure. Cultures were incubated in the dark at 26 °C.

ELSs that developed to a size >1 mm were transferred to KBP4P for the first two weeks, and later to K4NB, following the procedure described by Kumlehn et al. [81]. Cultures were maintained at 26 °C, with a photoperiod of 16/8 h (day/night) in dim light [80–100 μmol (hν) m^−2^ s^−1^ (PAR)]. The regenerated plantlets were transferred to MS medium without phytohormones, and subsequently planted into a mixture of soil and sand (3/1; *v*/*v*) once the root system had developed.

Six to ten wheat spikes were used for each isolation procedure (biological replication). At least three biological replications were made for all wheat lines after each tiller pre-treatment.

### 4.6. Cytological Analyses

Microspore viability (the percentage of fully viable microspores in the total population of isolated cells) was assessed immediately after isolation (0 d) based on the reaction of microspores with fluorescein diacetate (FDA; 0.01%; λ_Ex_ = 465 nm, λ_Em_ = 515 nm, green fluorescence), according to a method described by Heslop-Harrison and Heslop-Harrison [82]. Microspore samples were examined using a Nikon Eclipse E600 epifluorescence microscope (Nikon Instruments Inc., Tokyo, Japan) equipped with a Zyla 4.2 (Andor) camera (Andor Technology Ltd., Belfast, UK) and NIS-Elements AR 4.00 software. A minimum of 500 microspores from 10 fields of view (magnification × 100) were analyzed for each individual sample.

Isolated microspore cultures were examined by light microscopy using an Eclipse TS100 inverted microscope (Nikon, Tokyo, Japan) at the end of the isolation procedure (0 d), and after two (14 d) and four (28 d) weeks of in vitro culture.

### 4.7. Anther Culture

The stage of microspore development was analyzed in anthers collected from the central part of a spike, and stained with acetocarmine according to Barnabás et al. [83]. Only spikes containing microspores from the mid- or late-uninucleate stage were used in the study. Pre-treated spikes were sterilized for 4 min with 4.85% sodium hypochlorite (NaClO) solution for 4 min, and subsequently rinsed three times with sterile distilled water for 5 min. The anthers were then manually isolated from the central part of the spikes, and placed on 50 mm Petri dishes (50 anthers/dish) containing C17 induction medium [84] with 90 g/dm^3^ maltose, 1 mg/dm^3^ 2,4-D and 1 mg/dm^3^ Dicamba. The medium was solidified with 2.5 g/dm^3^ Phytagel or supplemented with 50 g/dm^3^ Ficoll 400.

The dishes were stored at 28 °C in the dark. Developed ELSs were transferred to MS regeneration medium [85] containing 0.5 mg/dm^3^ NAA and 0.5 mg/dm^3^ kinetin, and stored at 25 °C with a 16/8 h (light/dark) photoperiod. The regenerated plants were transferred onto MS medium without phytohormones, and then planted into the soil once the root system had developed.

For both isolated microspore and anther cultures, the effectiveness of ME induction was expressed as the number of ELSs developed and the number of GRs calculated per spike of the donor plant [ELS/spike; GR/spike].

### 4.8. Sampling for Biochemical Analyses

Anthers of three wheat lines (K20290, SM IHAR and PO19) were collected from freshly cut tillers, tillers pre-treated with low temperature (3–4 weeks at 4 °C) and tillers following a modified pre-treatment with low temperature, SSe and mannitol (3–4 weeks at 4 °C with the last 3 days in a solution of 50 µmol/dm^3^ sodium selenate followed by 4 days in 0.7 mol/dm^3^ mannitol at 20 °C). Each sample contained anthers randomly collected from approx. 30 spikes. Samples were immediately frozen in liquid N_2_ and stored at −60 °C.

### 4.9. Macro- and Microelement Analyses

Macro- and micronutrients were analyzed in wheat anthers using inductively coupled plasma-atomic emission spectroscopy (ICP-AES) and inductively coupled plasma mass spectrometry (ICP MS) according to procedures described by Zembala et al. [86] and Tobiasz et al. [74].

Lyophilized anthers (0.01 to 0.02 g) underwent digestion in a closed microwave system (Uni Clever, Plazmatronika, Poland) using 5 mL of ultrapure concentrated nitric acid (Merck, Darmstadt, Germany). The digests were diluted to 25 mL with deionized water. Macronutrients were determined using an ICP-AES spectrometer (Optima 2100, Perkin Elmer, Shelton, CT, USA) at the following wavelengths: K (766.5 nm), Ca (317.9 nm), Mg (285.2 nm) and Na (589.6 nm). Micronutrients were determined in relation to the following isotopes: Mn(55), Fe(57), Cu(63) and Mo(98) using an ICP-MS spectrometer (Elan DRC-e, Perkin Elmer, Shelton, CT, USA). Calibration was performed using multi-element standard sets (Perkin Elmer, Shelton, CT, USA).

### 4.10. Total Low-Molecular-Weight Antioxidant Activity

Samples were lyophilized under vacuum (0.070 mbar for 60 h) using a lyophilizer (Labconco UR). Subsequently, bulk samples of wheat anthers (about 0.3–0.5 g DW) were prepared, from which three representative samples (0.01–0.015 mg DW) were taken for analysis. The analytical procedure was described in detail in a previous study by Żur et al. [13].

Total antioxidant content, indicative of free radical scavenging activity, was measured using a 0.5 mmol/dm^3^ solution of stable free radical 1,1-diphenyl-2-picrylhydrazyl (DPPH, SIGMA, Munich, Germany) in methanol, following the method of Brand-Williams et al. [87] with some modifications, adapting the protocol to 96-well microtiter plates [10].

Each sample was measured in triplicate on separate plates by pipetting 100 µL of supernatant with the addition of 250 µL of 0.5 mmol/dm^3^ DPPH solution into 3 wells for each measurement. After incubating the reaction for 30 min at 37 °C, the absorbance was determined at 515 nm using a Model 680 microplate reader (Bio-Rad Laboratories, Hercules, CA, USA). The results were expressed as µmoles of Trolox equivalents g^−1^ DW, calculated using the linear regression equation from a calibration curve representing linear relationships between absorbance and Trolox concentrations (0.75, 1, 1.25, 1.5, 1.75, 2, 2.5 and 3 nmol/dm^3^). At least four measurements were made for two independent samples.

### 4.11. Sampling and Detecting Ascorbate and Glutathione

Anthers were ground with 6% meta-phosphoric acid (1:5 *w*/*v* ratio) in liquid N_2_ in a mortar until thawed. After centrifugation (15 min, 12,000× *g*, 4 °C), the supernatants were used for analyses. The levels of reduced and oxidized forms of ascorbate and glutathione in anther extracts were determined spectrophotometrically using an Ultrospec 2100 pro UV/visible spectrophotometer (Amersham, Umeå, Sweden). At least three independent measurements were carried out for all assays.

### 4.12. Reduced and Oxidized Glutathione Assay

Reduced (GSH) and oxidized (GSSG) glutathione were determined by an enzymatic recycling assay using glutathione reductase (GR) according to the method of Law et al. [88], which is described in detail in Żur et al. [10]. Briefly, the samples (50 µL of metaphosphoric acid extracts) were neutralized with 9 µL of 1.5 mol/dm^3^ triethanolamine (TEA). The total glutathione content was measured in an assay mixture containing 59 µL of neutralized sample, 50 mmol/dm^3^ potassium phosphate buffer (pH 7.4), 2.5 mmol/dm^3^ EDTA-Na_2_, 1 mmol/dm^3^ 5,5′-dithio-bis(2-nitrobenzoic acid) (DNTB), 0.6 U of GR from baker’s yeast and 0.2 mmol/dm^3^ NADPH in a total volume of 600 µL. The total glutathione content was estimated by measuring the absorbance at 412 nm.

GSSG measurements were carried out as for total glutathione, except that GSH was derivatized by adding 2 µL of 2-vinylpyridine to the neutralized samples and incubating for 1 h at room temperature. Total glutathione and GSSG contents were estimated from the standard curves based on a dilution series of GSH and GSSG in 6% metaphosphoric acid. GSH was calculated as the difference between total glutathione and GSSG concentrations.

### 4.13. Reduced and Oxidized Ascorbate Assay

The concentration of reduced ascorbate (ASC) was determined using the method of Foyer et al. [89] as modified by Harrach et al. [90]. The samples (62.5 µL of metaphosphoric acid extracts) were neutralized with 7.5 µL of 1.5 M TEA and mixed with 75 µL of 150 mM sodium phosphate buffer (pH 7.4) and 500 µL of 100 mM sodium phosphate buffer (pH 5.6). To oxidize ASC, 1 U of ascorbate oxidase (AO) from Cucurbita sp. (Sigma-Aldrich, Saint Louis, MO, USA) was added to the assay mixture. The ASC content was determined from the difference between the initial and final absorbance measured at 265 nm. Total ascorbate levels were determined after the reduction of dehydroascorbate (DHA) to ascorbate using dithiothreitol (DTT). A 10 mmol/dm^3^ DTT solution (37.5 µL) was added to the reaction mixture, prior to the addition of sodium phosphate buffer (pH 5.6) and AO, and incubated for 15 min at room temperature for 1 h.

Ascorbate concentration was calculated based on the extinction coefficient of ascorbic acid (14.7 mmol/dm^3^/cm). DHA content was obtained from the difference between total and reduced ascorbate concentrations.

### 4.14. Hydrogene Peroxide (H_2_O_2_)

The level of H_2_O_2_ generation was measured colorimetrically using the Amplex Red Hydrogen Peroxide Assay Kit (Invitrogen, Ontario, CA, USA), according to the manufacturer’s protocol, with some modifications. Subsequently, samples were homogenized in PBS buffer (1:4 *w*/*v* ratio) and liquid N_2_ in a mortar until thawed, and then centrifuged for 15 min at 12,000× *g*. The supernatants (50 μL) were then transferred to 96-well plates and mixed with a solution consisting of 100 mM Amplex Red reagent and 50 μL 0.2 U/mL horseradish peroxidase. After incubation for 30 min at room temperature, absorbance was read at 560 nm (Synergy II, Biotek, Winooski, VT, USA). H_2_O_2_ content was estimated from the standard curve prepared for the concentration range of 0.05–1 μmol/dm^3^.

## 5. Conclusions

In summary, all the data presented in this study lead to the conclusion that the high recalcitrance of Polish winter wheat breeding lines is associated with low stress tolerance at the developmental stage, optimal for ME induction. Stress associated with isolation and transfer to in vitro culture disrupts redox homeostasis and mineral nutrient management, resulting in excessive ROS generation, which drastically reduces microspore viability. Not only is a low-temperature treatment not sufficiently effective to reduce stress, but it also fails to completely arrest gametophytic development. Among various modifications, the procedure combining low temperature with the SSe and mannitol treatment has shown promise by increasing the total low-molecular-weight (LMW) antioxidant activity, effectively scavenging hydrogen peroxide. Overall, this procedure resulted in a moderate microspore yield and high microspore viability even in the poorly responsive genotypes. Although several parameters, such as the duration of the low-temperature tiller pre-treatment, SSe concentration or optimal temperature during mannitol application, as well as the in vitro culture conditions, still need optimization, the procedure shows high potential for the successful induction of microspore embryogenesis.

## Figures and Tables

**Figure 1 plants-13-00363-f001:**
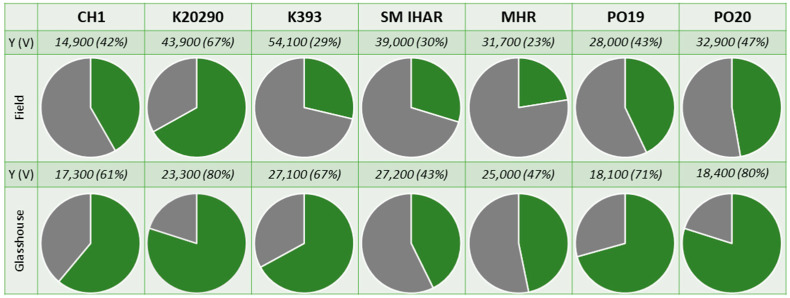
Microspore yield [mean number of microspores per spike] and viability [%] obtained for winter wheat breeding lines grown in the field and greenhouse conditions. Experiment 2020/2021. This description also applies to other figures in this paper. Left: the type of donor plant growth conditions. Above: the line. Pie charts—graphs divided into slices to illustrate the numerical proportion of viable microspores (green), dead microspores (grey). Above each chart are parameters of microspore yield (Y; mean number of microspores per spike) and microspore viability (V; (%)).

**Figure 2 plants-13-00363-f002:**
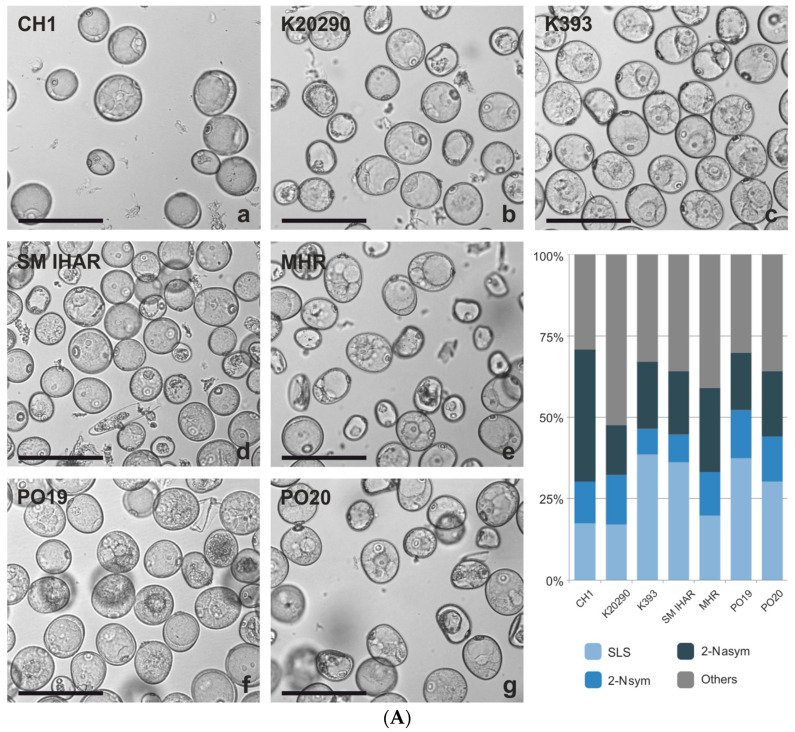
(**A**) Microspore suspension of the winter wheat lines (CH1, K20290, K393, SM IHAR, MHR, PO19, PO20) isolated from donor plants growing in the greenhouse on the day of isolation, and their cytological characteristics. (**B**) Microspores of the winter wheat lines (CH1, K20290, K393, SM IHAR, MHR, PO19, PO20) isolated from donor plants growing in the field on the day of isolation, and their cytological characteristics. Experiment 2020/2021. (**a**–**g**) Suspension with microspores showing typical features of developmental reprogramming: SLSs—star-like structures, 2-Nsym—structures after the first symmetric division, 2-Nasym—microspores continuing gametophytic development after the first asymmetric division, others—dead, damaged or unidentified microspores. Bar  =  100 µm; the graph shows the percentage of potentially embryogenic microspores.

**Figure 3 plants-13-00363-f003:**
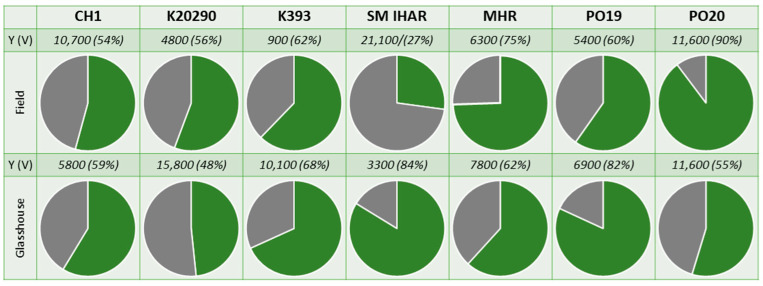
Microspore yield [mean number of microspores per spike] and viability [%] obtained for plants of winter wheat breeding lines grown under field or controlled conditions. Experiment 2021/2022. Left: the type of donor plant growth conditions. Above: the line. Pie charts—graphs divided into slices to illustrate the numerical proportion of viable microspores (green), dead microspores (grey). Above each chart are parameters of microspore yield (Y; mean number of microspores per spike) and microspore viability (V; (%)).

**Figure 4 plants-13-00363-f004:**
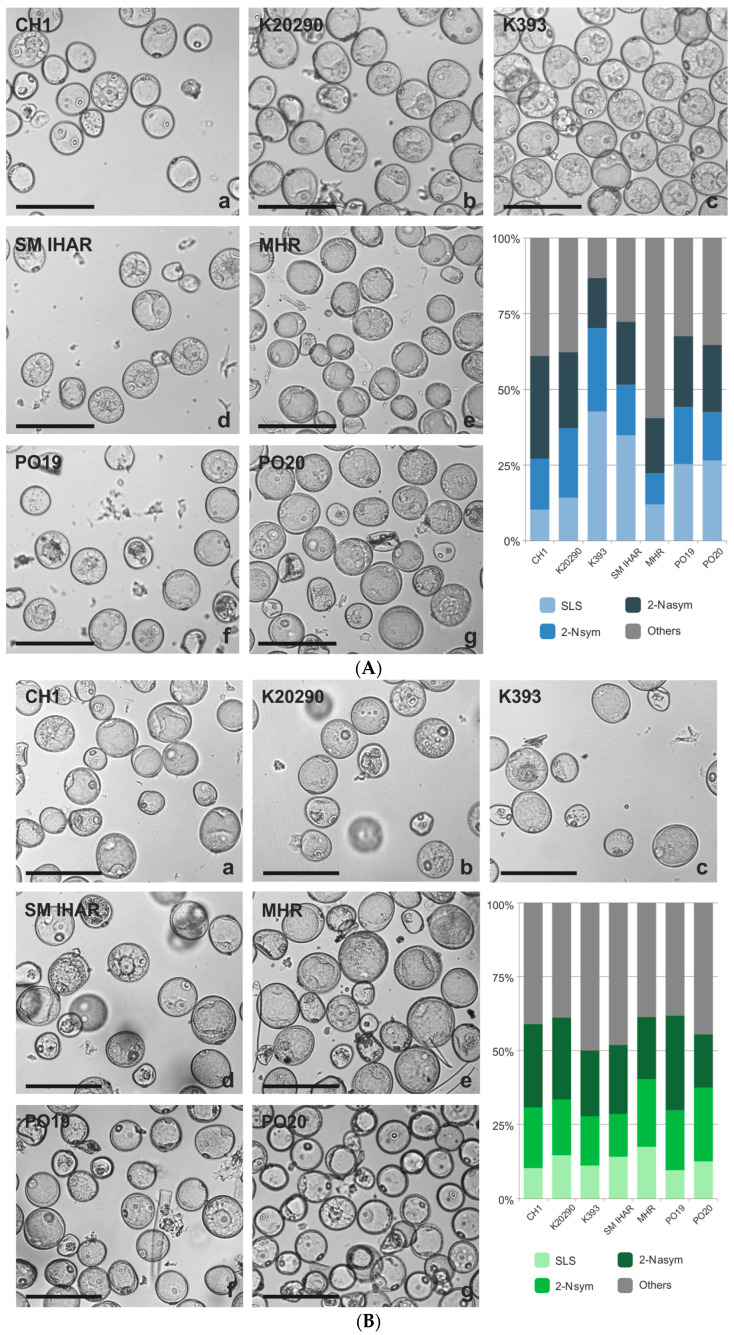
(**A**) Microspores of the winter wheat lines (CH1, K20290, K393, SM IHAR, MHR, PO19, PO20) isolated from donor plants growing in the greenhouse on the day of isolation and their cytological characteristics. (**B**) Microspores of the winter wheat lines (CH1, K20290, K393, SM IHAR, MHR, PO19, PO20) isolated from donor plants growing in the field on the day of isolation and their cytological characteristics. Experiment 2021/2022. (**a**–**g**) Suspension with microspores showing typical features of developmental reprogramming: SLSs—star-like structures, 2-Nsym—structures after the first symmetric division, 2-Nasym—microspores continuing gametophytic development after the first asymmetric division, others—dead, damaged or unidentified microspores. Bar  =  100 µm; the graph shows the percentage of potentially embryogenic microspores.

**Figure 5 plants-13-00363-f005:**
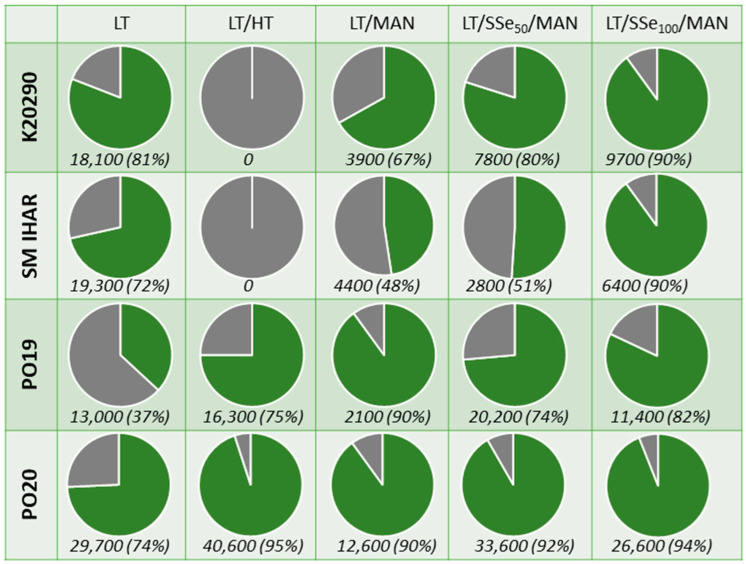
Microspore yield [mean number of microspores per spike] and viability [%] obtained for four selected winter wheat lines after various modified tiller pre-treatments. Experiment 2021/2022. Figure description. Left: the line. Above: the type of pre-treatment. Pie charts—graphs divided into slices to illustrate the numerical proportion of viable microspores (green), dead microspores (grey). Below each chart are parameters of microspore yield (Y; mean number of microspores per spike) and microspore viability (V; (%)). LT—Low-temperature tiller pre-treatment, 21–28 days at 4 °C; LT/HT—Low/high-temperature tiller pre-treatment: 21–28 days at 4 °C followed by 24 h at 30 °C; LT/MAN—Low-temperature/osmotic/starvation tiller pre-treatment: 21–28 days at 4 °C followed by 4 days in 0.4 mol/dm^3^ mannitol at 20 °C; LT/SSe50/MAN, LT/SSe100/MAN—Low-temperature/osmotic/starvation/selenium tiller pre-treatment: 21–28 days at 4 °C with the last three days in 50 µmol/dm^3^ or 100 µmol/dm^3^ sodium selenate solution (SSe50 and SSe100, respectively) followed by 4 days in 0.7 mol/dm^3^ mannitol at 20 °C.

**Figure 6 plants-13-00363-f006:**
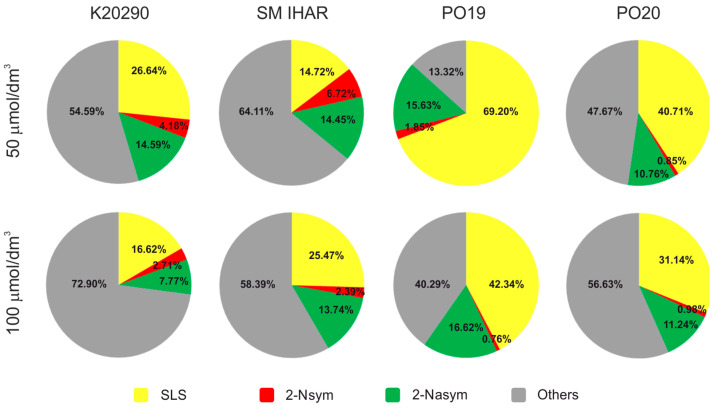
Cytological characterization of microspore suspensions from selected winter wheat lines isolated after a modified tiller pre-treatment combining low temperature (21–28 days at 4 °C) with application of sodium selenate (50 µmol/dm^3^ or 100 µmol/dm^3^ SSe for 3 days at 4 °C) and 0.7 mol/dm^3^ mannitol for 4 days at 20 °C. Experiment 2021/2022. Color gradients represent star-like structures (SLS); structures after the first symmetric division (2-Nsym); structures after the first asymmetric division 2-Nasym); others—dead, damaged or unidentified microspores.

**Figure 7 plants-13-00363-f007:**
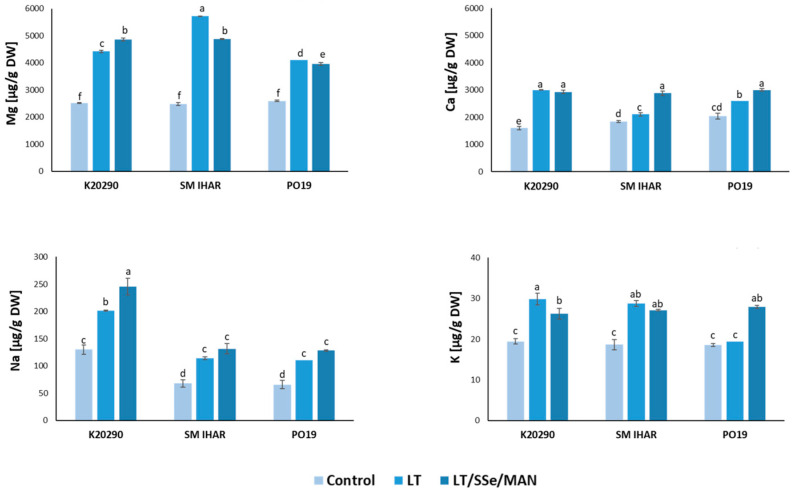
Content of selected macroelements (Mg, Ca, Na, K) and their changes induced by the ME induction procedure in anthers of three winter wheat lines (K20290, SM IHAR, PO19). Data represent ± SE. Values marked with the same letter do not differ significantly according to Duncan’s multiple range test (*p* ≤ 0.05). Control—anthers collected from non-pre-treated tillers; LT—anthers collected from low-temperature pre-treated tillers (21–28 days at 4 °C); LT/SSe/MAN—anthers collected from tillers pre-treated with low temperature (21–28 days at 4 °C), 50 µmol/dm^3^ sodium selenate (3 days at 4 °C) and 0.7 mol/dm^3^ mannitol (4 days at 20 °C).

**Figure 8 plants-13-00363-f008:**
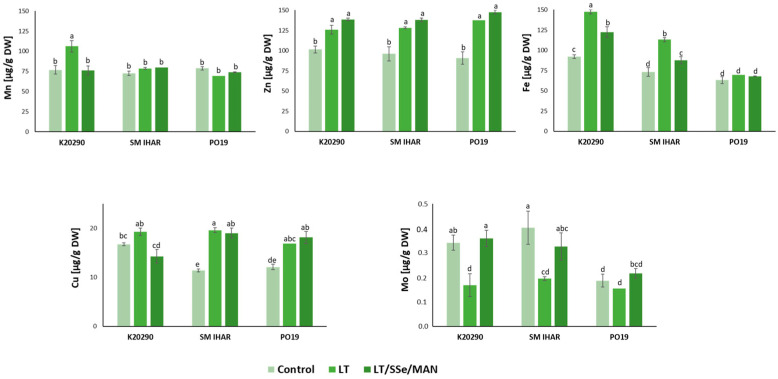
Content of selected microelements (Mn, Zn, Fe, Cu, Mo) and their changes induced by the ME induction procedure in anthers of three winter wheat lines (K20290, SM IHAR, PO19). Data represent means ± SE. Values marked with the same letter do not differ significantly according to Duncan’s multiple range test (*p* ≤ 0.05). Control—anthers collected from non-pre-treated tillers; LT—anthers collected from low-temperature pre-treated tillers (21–28 days at 4 °C); LT/SSe/MAN—anthers collected from tillers pre-treated with low temperature (21–28 days at 4 °C), 50 µmol/dm^3^ sodium selenate (3 days at 4 °C) and 0.7 mol/dm^3^ mannitol (4 days at 20 °C).

**Figure 9 plants-13-00363-f009:**
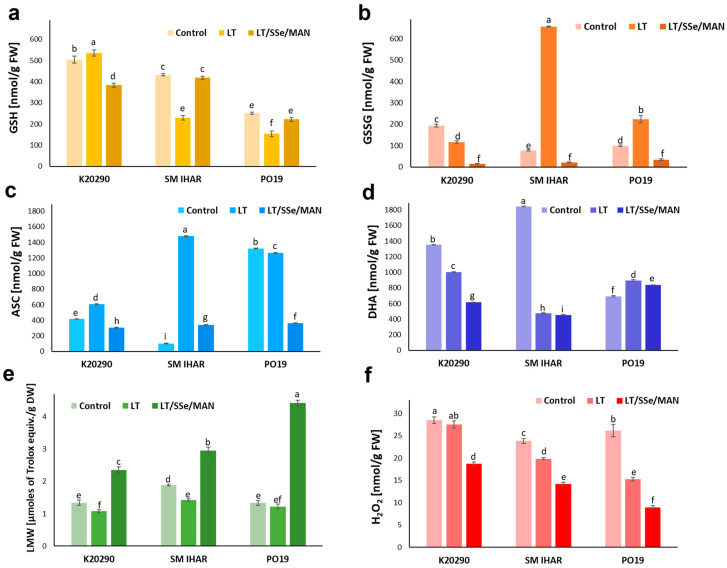
Antioxidative defense provided by low-molecular-weight antioxidants and hydrogen peroxide level in the anthers of three winter wheat lines (K20290, SM IHAR, PO19) and their changes induced by ME induction procedure. Data represent means ± SE. Values marked with the same letter do not differ significantly according to Duncan’s multiple range test (*p* ≤ 0.05). Content of (**a**) reduced glutathione (GSH); (**b**) oxidized glutathione (GSSG); (**c**) reduced ascorbate (ASC); (**d**) oxidized ascorbate (DHA); (**e**) total low-molecular-weight antioxidants activity (the DPPH radical scavenging activity expressed as Trolox equivalent antioxidant capacity); (**f**) level of hydrogen peroxide (H_2_O_2_). Control—anthers collected from non-pre-treated tillers; LT—anthers collected from tillers pre-treated with low temperature (21–28 days at 4 °C); LT/SSe/MAN—anthers collected from tillers pre-treated with low temperature (21–28 days at 4 °C), 50 µmol/dm^3^ sodium selenate (3 days at 4 °C) and 0.7 mol/dm^3^ mannitol (4 days at 20 °C).

**Table 1 plants-13-00363-t001:** Levels of total glutathione and ascorbate (GSH + GSSG; ASC + DHA; [nmol/g FW]) and the shares of their reduced forms in the total pools measured in the anthers of three winter wheat lines (K20290, SM IHAR, PO19) during the ME induction procedure. Data represent means ± SE. Values marked with the same letter do not differ significantly according to Duncan’s multiple range test (*p* ≤ 0.05).

Wheat Line	Treatment	GSH + GSSG	GSH	ASC + DHA	ASC
			GSH + GSSG		ASC + DHA
K20290	Control	697.5 ± 12.3 ^b^	0.72	1770.7 ± 3.4 ^d^	0.24
	LT	651.7 ± 8.8 ^c^	0.82	1608.2 ± 5.9 ^e^	0.38
	LT/SSe/MAN	398.6 ± 9.8 ^f^	0.96	924.3 ± 3.9 ^g^	0.33
SM IHAR	Control	510.4 ± 4.3 ^d^	0.85	1946.7 ± 5.9 ^c^	0.05
	LT	885.1 ± 11.7 ^a^	0.26	1955.8 ± 3.4 ^c^	0.76
	LT/SSe/MAN	440.8 ± 7.0 ^e^	0.95	797.5 ± 3.8 ^h^	0.43
PO19	Control	350.2 ± 3.0 ^g^	0.72	2014.4 ± 3.9 ^b^	0.66
	LT	377.2 ± 3.6 ^f^	0.41	2161.1 ± 12.0 ^a^	0.59
	LT/SSe/MAN	256.2 ± 5.4 ^h^	0.87	1199.6 ± 5.9 ^f^	0.30

Control—anthers collected from non-pre-treated tillers; LT—anthers collected from low-temperature pre-treated tillers (21–28 days at 4 °C); LT/SSe/MAN—anthers collected from tillers pre-treated with low temperature (21–28 days at 4 °C), 50 µmol/dm^3^ sodium selenate (3 days at 4 °C) and 0.7 mol/dm^3^ mannitol (4 days at 20 °C); GSH, ASC—reduced forms of glutathione and ascorbate; GSSG—glutathione disulphide, oxidized form of glutathione; DHA—dehydroascorbic acid; oxidized form of ascorbate.

## Data Availability

Data are contained within the article and Appendix A.

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
