# Peer review of "New Prospects for Improving Microspore Embryogenesis Induction in Highly Recalcitrant Winter Wheat Lines"

_plants, 2024, doi:10.3390/plants13030363_

Round 1
Reviewer 1 Report
Comments and Suggestions for Authors
The paper generally does not clearly describe the results of a comprehensive discussion. An example can be taken from Figure 3, in which legends and text present a very bad description. The authors use colors in the graphs and the letters Y (yield) and V (viability) what are the colors for each of these Y or V? Also the text is not clear. Continuing the text and corresponding figures the authors should identify in the microscopic images what microscopes are undergoing asymmetric divisions. The results reported present a different behavior from micropores from in-field growing plants and greenhouse-growing plants. The authors have no comments on these differences and/or possible reasons. In the text and figures 3 and 4, the authors refer to the frequency which is not correct if they are reporting on the data of the figures. In figures they present percentages and in the legends and text they report on frequency which is not at all the same. Continuing to read the text and comparing with figures we can take as an example fig 5. In the graph, the authors want to represent microspore yield (mean number of microspores/spike) and viability. They use two colors. What does each color? The legends in general are not comprehensive and in consequence, the description of the results follows the same criticism. The paper is not well revised by the authors and along with all the text and figures the comments are similar. Again the description of the results of Fig. 6 the reader is confronted with misleading text. The authors report on color gradients. From my eyes, there are two different colors (blue and green)and not a color as written. In lines 330 to 332 the authors state that pre-treatment decreased or decreased Ng and K levels. The authors have to be rigorous in describing the results referring if the differences are significant or not. Also since the authors in the graphs refer to a/g/DW they can't refer to levels in the text. In general, the presentation of the results is not rigorous. In the discussion, the authors introduce literature on anther culture in particular from lines 547 to 557. However, the paper concerns microspore embryogenesis and not anther embryogenesis. There is no reason to discuss it in that way. It was interesting to present results on the evaluation of different elements. However, the results obtained were not discussed in terms of their role in ME induction wich should constitute a novelty of the study. At the end of the paper the reader is not informed on the advantage/disadvantage of using in-field or greenhouse-growing plants. It is not clear if the authors were able to produce embryos from the microspores of the different varieties used. Should the authors be able to propose a protocol for the obtention of microspore-derived embryos of what variety? From my point of view, this paper needs a strong revision to make it important in scientific terms and for agricultural purposes.
Comments on the Quality of English Language
The English has to be reviewed by an English mother language scientist or reviewer
Author Response
Responses to R1:
Comment 1: The paper generally does not clearly describe the results of a comprehensive discussion. An example can be taken from Figure 3, in which legends and text present a very bad description. The authors use colors in the graphs and the letters Y (yield) and V (viability) what are the colors for each of these Y or V? Also the text is not clear. Continuing the text and corresponding figures the authors should identify in the microscopic images what microscopes are undergoing asymmetric divisions.
Response 1: Thank you for pointing this out. We agree with this comment. Therefore, we have improved Figures 1,3, 5 and their description, both in the legend and the text.
Comment 2: The results reported present a different behavior from micropores from in-field growing plants and greenhouse-growing plants. The authors have no comments on these differences and/or possible reasons.
Response 2: Polish winter wheat breeding materials are highly recalcitrant to microspore embryogenesis induction, both in anther and isolated microspore cultures. The main aim of our studies was to identify the possible causes and the mechanism leading to this failure in microspore reprogramming. Therefore, we started our studies with the standard, simple procedures highly recommended by other authors working on the process of microspore embryogenesis. We introduced various modifications, starting from very simple variables such as the growing conditions of the donor plants. Due to the large amount of data and the limited size of the manuscript, we have not discussed all of them in detail, but have concentrated on the new, original points.
It is well known that plants grown in natural environment have higher vigour and better physiological condition (produce more tillers with larger spikes, more anthers and higher number of microspores) than plants grown in growth chambers. This could be the result of mycorrhizal symbiosis, but this aspect of microspore embryogenesis should first be studied in detail. On the other hand, this positive effect could be drastically reduced by high temperature, drought, pathogenic infections or other unfavourable environmental conditions that reduce microspore viability. Most reports recommend controlled growth conditions for donor plants, and in the first replication of our experiment, the mean microspore viability of five breeding lines of the F1 generation was 22% higher in plants grown under controlled conditions. It seems quite obvious that this difference will fluctuate (in the next experimental replications it was not significant).
Comment 3: In the text and figures 3 and 4, the authors refer to the frequency which is not correct if they are reporting on the data of the figures. In figures they present percentages and in the legends and text they report on frequency which is not at all the same. Continuing to read the text and comparing with figures we can take as an example fig 5. In the graph, the authors want to represent microspore yield (mean number of microspores/spike) and viability. They use two colors. What does each color? The legends in general are not comprehensive and in consequence, the description of the results follows the same criticism.
Response 3: Thank you for pointing this out. We agree partially with this comment.
In agreement with response 1 we have improved Figure 3 and its description, in both the legend and the text.
We were using term frequency expressed as a percentage [%]. Microspore frequencies can be expressed as a percentage or as a fraction. In a population, microspore frequencies reflect specific diversity (e.g. Fig. 3 - dead versus vital or Figs. 4A-B. embryogenic versus non-embryogenic).
However, as suggested, we have replaced the term frequency with percentage both in the figures (Figs. 2, 4, 6), in the legend and in the text.
Comment 4: The paper is not well revised by the authors and along with all the text and figures the comments are similar. Again the description of the results of Fig. 6 the reader is confronted with misleading text. The authors report on color gradients. From my eyes, there are two different colors (blue and green)and not a color as written.
Response 4: Thank you for bringing this to our attention. We agree with this comment.
The colours used in Figure 6 have been changed to provide more contrast. We apologise for the error in citing Figure 6 in the text. The error in figure numbering (Fig. 5 instead of Fig. 6 on lines 268, 274, and 277) has been corrected.
Comment 5: In lines 330 to 332 the authors state that pre-treatment decreased or decreased Ng and K levels. The authors have to be rigorous in describing the results referring if the differences are significant or not. Also since the authors in the graphs refer to a/g/DW they can't refer to levels in the text. In general, the presentation of the results is not rigorous.
Response 5: Thank you for bringing this to our attention. We agree partially with this comment.
In lines 330 to 332 we stated that pre-treatment increased or decreased Mg and K levels compared to the to the effect of low temperature. We described these changes as significant (according to Duncan’s multiple range test, see legend). On this basis, increased or unchanged means not significant - as was described for Na and Ca levels in line 332.
We have replaced the text: “Compared to the effect of low temperature, the combined low temperature, SSe and mannitol pre-treatment increased or decreased Mg and K levels, and elevated or did not change Na and Ca levels (Figure 7)” with a more precise description: “Compared to the effect of low temperature, the combined low temperature, SSe and mannitol pre-treatment increased (K20290) or decreased (SM IHAR, PO19) Mg content, increased (PO19) or decreased (K20290) K content, elevated (K20290) or did not change (SM IHAR, PO19) Na content and elevated (SM IHAR, PO19) or did not change (K20290) Ca content (Figure 7).”
In Figure 8 we referred to the content expressed as [µg/g DW]. We have replaced "level" with "content" in the text: lines 321, 323, 324, 332, 348, 350, 353, 355, 356, 359-361.
Comment 6: In the discussion, the authors introduce literature on anther culture in particular from lines 547 to 557. However, the paper concerns microspore embryogenesis and not anther embryogenesis. There is no reason to discuss it in that way.
Response 6: Thank you for bringing this to our attention. We agree partially with this comment.
We presented both anther and microspore cultures as techniques used to induce microspore embryogenesis. Consequently, we discussed our results with other authors on anther and microspore cultures from lines 547 to 557. This fragment related to the tapetum (present only in anthers) affecting microspores viability and potential to form proper developed ELS.
Comment 7: It was interesting to present results on the evaluation of different elements. However, the results obtained were not discussed in terms of their role in ME induction which should constitute a novelty of the study.
Response 7: We discussed the results of the evaluation of different elements in lines: 601-615 and 646-666. We point out the role of Mo, as the element possibly related to the induction potential of microspore embryogenesis.
Comment 8: At the end of the paper the reader is not informed on the advantage/disadvantage of using in-field or greenhouse-growing plants. It is not clear if the authors were able to produce embryos from the microspores of the different varieties used. Should the authors be able to propose a protocol for the obtention of microspore-derived embryos of what variety? From my point of view, this paper needs a strong revision to make it important in scientific terms and for agricultural purposes.
Response 8: The aspect of the effect of field versus controlled growth conditions is discussed in lines 512-521.
All the results showing effectiveness of ELS production and plant regeneration are presented in Tables1-4. As stated at the beginning of the discussion: “This research sheds new light on the possibilities of incorporating DH technology in wheat breeding. While confirming the high recalcitrance of Polish winter wheat lines to ME induction, the study also explored genotypes with potentially higher responsiveness (PO19, PO20), selected on the basis of previous studies”. The aim of this study was to identify the main causes of such high recalcitrance. We have developed a procedure that is potentially effective in microspore reprogramming which is the first step for effective DH technology. However, we have not yet overcome the barrier of further correct development of multicellular structures, which will be the next step in our studies. We have identified the main causes of low responsiveness of the wheat lines studied, which is low tolerance to applied stress treatment, and demonstrated the beneficial effect of mannitol and selenium on both microspore vitality and embryogenic potential.
All these data give us important information on the nature of the problems observed and can be used to start developing effective protocols.

Reviewer 2 Report
Comments and Suggestions for Authors The main idea of anther/microspore culture is to develop haploids for doubled haploid breeding. Anther/microspore response is highly genotypic dependent, which limits their application in different genotypes for developing new varieties. Cereals have been recalcitrant in producing embryos. Comments - increase the bar size of specific pictures highlighting results - provide pictures of microspore embryos, germination -provide histological evidence of embryogenesis -show pictures of viable and non-viable microspores -germination rate of embryos -the quality of embryos -the percentage of in vitro embryo-derived plants - show variation in ploidy level, if any Earlier, plenty of work has been done in microspore cultures- low/high-temperature treatment, and mannitol treatment in many crops for enhancing microspore response for embryo production. Therefore, it would be useful to study stress-related protein and molecular-level changes

Author Response
Responses to R2:
Comments 1: Increase bar size of specific pictures highlighting results.
Response 1: Thank you for pointing this out. We agree with this comment. Therefore, we have increased the bar size to 100 µm in Figures 2 A-B and 4 A-B.
Comments 2: Provide pictures of microspore embryos.
Response 2: The images of microspore-derived embryos have been provided in Supplementary Figs. S2 C-D, showing properly developed embryo-like structures (ELS) at the globular stage after 14 and 42 days of in vitro culture.
Comments 3: Provide histological evidence of embryogenesis.
Response 3: Polish winter wheat breeding materials are highly recalcitrant to microspore embryogenesis induction, both in anther and isolated microspore cultures. The main aim of our studies was to identify the possible causes and the mechanism leading to this failure in microspore reprogramming. This manuscript presents the first part of the received results obtained, confirming the high recalcitrance of the wheat lines studied and identifying the main problems: low microspores viability and some disturbances in ELS development. Due to the current limitations of the manuscript, further data from the study will be reported in future publications. Due to the low number of microspore-derived ELS, all were transferred to the regeneration medium to determine its regeneration potential.
To date, many studies on various plant species have precisely described histological evidence of an ELS development in both microspore and anther cultures (Dubas et al. 2014; 2021). Generally, there are always both ELS with a proper elongation axis and various failure embryo phenotypes. Only correct haploid ELS can subsequently germinate and grow into mature plants, so the regeneration ability (in our manuscript presented on supplementary Figures S1-S4 and Tables S1-S4 as GR per spike) reflects the frequency of ELS with proper phenotype (correct apical - basal axis).
Improving the effectiveness of microspore embryogenesis induction, which we hope will be the outcome of our study, will allow for the detailed histological characterisation. This is planned as the next step in our studies.
Dubas et al. (2014) Protoplasma 251: 1077-1087.
Dubas et al. (2021) BMC Plant Biol 21, 586.
Comments 4: Provide germination rate of embryos.
Response 4: Germination rate of embryos is expressed as ‘GR per spike’ and ‘GR per 100 anthers’ parameters. It is provided in supplementary Figures S1-S4 and Tables S1-S4.
Comments 5: Provide quality of embryos.
Response 5: The images of microspore-derived ELSs are presented in Supplementary Fig. S2, where the quality is highlighted (proper ELS on Figs. C-D; incorrect ELS on Figs: G - Aborted multicellular structures and H - Callus-like structure).
Improving the effectiveness of microspore embryogenesis induction, which we hope will be the outcome of our study, will allow for the detailed histological characterisation. This is planned as the next step in our studies.
Comments 6: Provide percent of in vitro embryo derived plants.
Response 6: We have already provided supplementary Figures S1-S4 focusing on the regeneration rate of green plants (GR per spike Tables S1, S3 or GR per 100 anthers Tables S2, S4).
Comments 7: Show any variation in ploidy level.
Response 7: Due to the small number of regenerated plants, the ploidy level could not be determined. However, both their morphology (small size, low vigour, narrow leaves) and the fact that none of the regenerated plants produced seeds, suggest that their ploidy level is incorrect (haploids or mixoploids). Spontaneously produced doubled haploids were not observed. Improving the efficiency of microspore embryogenesis induction, which we hope will be the result of our study, will allow detailed ploidy determination.
Comment 8: Where is the control? without any temperature treatment/osmotic stress for comparing results?
Response 8: Many years of research (Touraev et al. 1996; 1997) have shown that isolation of microspores/anthers without any stress pre-treatment is not possible, as the majority of microspores die during the isolation procedure, or during the first hours of in vitro culture. Stress has been shown to be a critical factor to induce microspore embryogenesis. Therefore, only controls (anthers collected from non-pretreated tillers) were used only for all biochemical analyses.
Touraev et al. (1996). Planta 200: 144-152.
Touraev et al. (1997). Trends Plant Sci 2:285-303.
Comment 9: Earlier, plenty of work has been done in microspore cultures- low/high temperature treatment, mannitol treatment in many crops for enhancing microspore response for embryo production. Therefore, it would be useful to study stress related protein and also molecular level changes.
Response 9: Exactly, the identification of stress-related protein and also molecular changes associated with wheat microspore embryogenesis are planned as the next step in our study. But first we need to improve the effectiveness of the process and select high quality plant material. We are getting closer to this step. Based on our expertise, also in proteomics, we have to work with at least two DH lines which differ significantly in their response to ME-inducing treatment.
The next step will be a comparative proteomic analysis, which will provide new insights into the regulation of the induction of microspore embryogenesis in wheat.

Round 2
Reviewer 1 Report
Comments and Suggestions for Authors
The paper is interesting but has a low originality.
The authors have identified compounds known and described for many aromas. In what concerns the relationship of the metabolites with the transcripts the authors refer to classes and the relationship with the metabolites differentially produced along the floral stages is unclear.
Reading the paper we can not find any application of the study to the potential uses of this Dendrobium sp.. A similar comment can be said for the conclusions that don't bring a clear idea of how the results reported can be used for exploring potential applications of the floral scent of Dendrobium devonianum in food products.
The discussion has to be revised to use the data obtained for the paper's aim.
The discussion has to be revised to use the data obtained for the paper's aim.
Comments on the Quality of English Language
The English needs minor revisions
Author Response
Responses to R1:
Comment 1: The paper is interesting but has a low originality.
Response 1: Polish winter wheat breeding materials are highly recalcitrant to microspore embryogenesis induction, both in anther and isolated microspore cultures. The main aim of our studies was to identify the possible causes and the mechanism leading to this failure in microspore reprogramming. This manuscript presents the first part of the received results obtained, confirming the high recalcitrance of the wheat lines studied and identifying the main problems: low microspores viability and some disturbances in ELS development.
The study of several wheat breeding lines allows us to create a model consisting of three selected winter wheat lines with different potential for ME induction, which will be used in further studies. We identified the main cause of the high sensitivity of microspores to the applied stress treatments and found the possibility of increasing their stress tolerance, resulting in higher viability and enhanced potential for embryogenic development. The beneficial effect of selenium on both microspore vitality and embryogenic potential is a complete novelty that has never been reported before.
Due to the current limitations of the manuscript, further data from the study will be reported in future publications, showing a comparative analysis of selected breeding lines, which will provide new insights into the regulation of the induction of microspore embryogenesis in wheat.
The article fits perfectly into a special issue of the journal Plants entitled "Plant Developmental Pathways: Haploid, Zygotic and Somatic Embryos" to which we would like to submit. We believe, it is original and can be of value to scientists and breeders working not only with wheat but also with other crops that are recalcitrant to ME induction.
Comment 2: The authors have identified compounds known and described for many aromas. In what concerns the relationship of the metabolites with the transcripts the authors refer to classes and the relationship with the metabolites differentially produced along the floral stages is unclear. Reading the paper we can not find any application of the study to the potential uses of this Dendrobium sp.. A similar comment can be said for the conclusions that don't bring a clear idea of how the results reported can be used for exploring potential applications of the floral scent of Dendrobium devonianum in food products.
The discussion has to be revised to use the data obtained for the paper's aim.
Response 2: We apologize, but on reading Comment 2, we cannot find any retaliation with our manuscript (Keywords: Dendrobium sp.; floral; food; metabolites; transcripts). Comment 2 requested additional data or analyses or discussions "beyond the scope" of the present work on wheat microspore embryogenesis (Keywords: microspore embryogenesis; winter wheat; macro- and micronutrients; stress; antioxidants; hydrogen peroxide).
Comment 3: Comments on the Quality of English Language - The English needs minor revisions.
Response 3: The quality of the English language of the text of our manuscript (Manuscript # plants-2768706) has been checked and corrected after professional proofreading (English certificate translator office; we have the official document of payment – invoice).

Reviewer 2 Report
Comments and Suggestions for Authors
The authors have responded to my queries and I am not satisfied with their response. It is premature to publish these results and resubmit them. They must include new results based on my suggestions.
Comments on the Quality of English LanguageThe authors have responded to my queries and I am not satisfied with their response. It is premature to publish these results and resubmit them. They must include new results based on my suggestions.
Author Response
Responses to R2:
Comment 1: The authors have responded to my queries and I am not satisfied with their response. It is premature to publish these results and resubmit them. They must include new results based on my suggestions.
Response 1: Polish winter wheat breeding materials are highly recalcitrant to microspore embryogenesis induction, both in anther and isolated microspore cultures. The main aim of our studies was to identify the possible causes and the mechanism leading to this failure in microspore reprogramming. This manuscript presents the first part of the received results obtained, confirming the high recalcitrance of the wheat lines studied and identifying the main problems: low microspores viability and some disturbances in ELS development.
The study of several wheat breeding lines allows us to create a model consisting of three selected winter wheat lines with different potential for ME induction, which will be used in further studies. We identified the main cause of the high sensitivity of microspores to the applied stress treatments and found the possibility of increasing their stress tolerance, resulting in higher viability and enhanced potential for embryogenic development. The beneficial effect of selenium on both microspore vitality and embryogenic potential is a complete novelty that has never been reported before.
Due to the current limitations of the manuscript, further data from the study will be reported in future publications, showing a comparative analysis of selected breeding lines, which will provide new insights into the regulation of the induction of microspore embryogenesis in wheat.
The article fits perfectly into a special issue of the journal Plants entitled "Plant Developmental Pathways: Haploid, Zygotic and Somatic Embryos" to which we would like to submit. We believe, it is original and can be of value to scientists and breeders working not only with wheat but also with other crops that are recalcitrant to ME induction.
We cannot imagine that the results obtained will not be of interest to scientists working on microspore embryogenesis in wheat breeding.

Round 3
Reviewer 2 Report
Comments and Suggestions for Authors
The authors have justified queries made earlier, even though more experiments must be carried out on histology, and stress-related proteins, high germination rate of microspore embryos, and ploidy level. Still, there is more work to be carried out before these results can be implemented for improving microspore embryogenesis by reducing the recalcitrance of wheat crops.